# Physicochemical Characterization of Chitosan/Poly-γ-Glutamic Acid Glass-like Materials

**DOI:** 10.3390/ijms241512495

**Published:** 2023-08-06

**Authors:** Sondos Hejazi, Odile Francesca Restaino, Mohammed Sabbah, Domenico Zannini, Rocco Di Girolamo, Angela Marotta, Sergio D’Ambrosio, Irene Russo Krauss, C. Valeria L. Giosafatto, Gabriella Santagata, Chiara Schiraldi, Raffaele Porta

**Affiliations:** 1Department of Chemical Sciences, University of Naples “Federico II”, 80126 Naples, Italy; sondosmohammadhasan.hejazi@unina.it (S.H.); odilefrancesca.restaino@unina.it (O.F.R.); or domenico.zannini@ipcb.cnr.it (D.Z.); rocco.digirolamo@unina.it (R.D.G.); irene.russokrauss@unina.it (I.R.K.); giosafat@unina.it (C.V.L.G.); 2Department of Nutrition and Food Technology, An-Najah National University, Nablus P400, Palestine; m.sabbah@najah.edu; 3Institute for Polymers, Composites, and Biomaterials, National Council of Research, 80078 Pozzuoli, Italy; gabriella.santagata@ipcb.cnr.it; 4Department of Chemical, Materials and Production Engineering (DICMaPI), University of Naples “Federico II”, 80126 Naples, Italy; angela.marotta@unina.it; 5Department of Experimental Medicine, Section of Biotechnology and Molecular Biology, University of Campania “Luigi Vanvitelli”, 80138 Naples, Italychiara.schiraldi@unicampania.it (C.S.); 6Consorzio per lo Sviluppo dei Sistemi a Grande Interfase, 50019 Florence, Italy

**Keywords:** chitosan, poly-γ-glutamic acid, polyelectrolytes complexation, crosslinked materials, glass-like materials, bioplastics, physicochemical characterization

## Abstract

This paper sets up a new route for producing non-covalently crosslinked bio-composites by blending poly-γ-glutamic acid (γ-PGA) of microbial origin and chitosan (CH) through poly-electrolyte complexation under specific experimental conditions. CH and two different molecular weight γ-PGA fractions have been blended at different mass ratios (1/9, 2/8 and 3/7) under acidic pH. The developed materials seemed to behave like moldable hydrogels with a soft rubbery consistency. However, after dehydration, they became exceedingly hard, glass-like materials completely insoluble in water and organic solvents. The native biopolymers and their blends underwent comprehensive structural, physicochemical, and thermal analyses. The study confirmed strong physical interactions between polysaccharide and polyamide chains, facilitated by electrostatic attraction and hydrogen bonding. The materials exhibited both crystalline and amorphous structures and demonstrated good thermal stability and degradability. Described as thermoplastic and saloplastic, these bio-composites offer vast opportunities in the realm of polyelectrolyte complexes (PECs). This unique combination of properties allowed the bio-composites to function as glass-like materials, making them highly versatile for potential applications in various fields. They hold potential for use in regenerative medicine, biomedical devices, food packaging, and 3D printing. Their environmentally friendly properties make them attractive candidates for sustainable material development in various industries.

## 1. Introduction

The rapid emergence of innovative technologies at reduced costs has created a matrix of potential challenges to find new sustainable alternatives to plastic pollution. Only one-fourth of fossil-based materials are recycled [1], making the turn to eco-sustainable, biodegradable polymers unavoidable. Bioplastics produced from renewable sources gained a huge interest as possible replacements for oil-based plastics [2]. In fact, the European Sustainable Development agenda aims to achieve a wider use of bioplastics by 2030 [3]. Traditional plastics are constituted by polymers of specific repeating units that determine their properties. These features, common not only to polyethylene, polypropylene, polystyrene, and polyvinyl chloride but also to biopolymers such as polylactic acid, starch, or cellulose derivatives, appear to confer to plastic materials key properties [4]. Despite several research efforts dedicated to biopolymers in recent years, little is known about the ability of polypeptides or homopolypeptides to give rise to materials with specific plastic-like properties [5]. Among these, poly-γ-glutamic acid (γ-PGA), a natural, anionic homopolyamide (molecular weight (Mw) from 20 to 2000 kDa), is made of D- and/or L-glutamic acid units that are linked by amide linkages between the α-amino group and γ-carboxylic acid [6]. It is widely used in medicine and food industry applications, and can potentially replace synthetic plastics, being both non-toxic and biodegradable [4,5]. γ-PGA is produced by *Bacillus* strain fermentation [6,7], but as a pure powder, it is expensive, according to the Merck price list (100 mg costs 291 €) [8], due to low production yields and complex purification procedures [9]. Cheaper bulks, having low γ-PGA content and pureness, are also present on the market but need to be purified [10] before being employed [10,11]. However, the use of food by-products offers a promising solution in terms of cost and sustainability, as shown by Hisada and Kawase (2017), who have explored the utilization of wasted natto mucilage as a fermentation source for industrial-scale production of γ-PGA [12]. Chitosan (CH), instead, is a polysaccharide made of β-1,4-linked glucosamine, partially acetylated (≤25%), derived from the chitin of shell shrimps [13]. CH is soluble in aqueous acidic solutions, where the amine groups are protonated [14], making it form polyelectrolyte complexes with highly negatively charged polymers, such as γ-PGA [15]. Thus, crosslinked materials were prepared in this paper by physical blending of γ-PGA and CH. Their physicochemical properties, thermal behavior, and degradability were evaluated and compared with those displayed by the native biopolymers. 

## 2. Results and Discussion

### 2.1. Poly-γ-Glutamic Acid and Chitosan Molecular Weight Determination

A commercial γ-PGA sample (COM-PGA) was employed, the pureness of which was 51.8 ± 0.5% on the dry weight (water content was 8.7 ± 0.2%), as analyzed by ultra-high performance liquid chromatography (UHPLC) [10]. It was ultrafiltered (UF) after being dissolved in MilliQ water (18.7 ± 0.8% of the sample remained insoluble) on 100 and 3 kDa cut-off membranes. The 100 kDa (R1) and 3 kDa (R2) retentates contained 8.7 ± 0.7% and 10.6 ± 0.5% of the initial γ-PGA amount with a purity of 72.4 ± 0.8% and 73.9 ± 0.5% on the dry weight, respectively, which was similar to previous results [10]. During the process, the mass recoveries of R1 plus the 100 kDa permeate (P1) and of R2 plus the 3 kDa permeate (P2) were 90.8% and 99.0%, respectively (Appendix A Table A1). The COM-PGA Mw distribution, determined by size-exclusion chromatography with triple detector array (SEC-TDA), showed three peaks of 77.4, 20.6, and 2.92 kDa average Mw. These populations were separated by UF in R1 and R2, which showed 59.4 and 20.1 kDa Mw with a representativity of 93.2% and 93.6%, respectively, and were then used to prepare the materials (Figure 1a,b). Previous data demonstrated that COM-PGA lowest Mw fraction (2.92 kDa) was not suitable to prepare materials [10]. Conversely, CH showed a main peak (75.0% of representativity) of about 285.5 kDa with a dispersity (Mw/Mn) of 1.6 (Figure 1a,b). 

### 2.2. Hydrophilicity and Solubility

Crosslinked materials were prepared by blending solutions of CH and of the two γ-PGA fractions (R1 and R2) at pH 3.5 and weight ratios of 1/9, 2/8, 3/7 CH/R1, or CH/R2. The precipitated materials separated first by decantation and then by centrifugation after several washings with the same solvent resulted in soft, rubbery, easy-to-manipulate materials that were moldable in a wet state. 

Astonishingly, once dehydrated at ambient temperature, they turned into extremely hard and glass-like materials. Comparable findings were published, describing rubbery hydrogels obtained by blending CH with carrageenan [16]. Since its rubbery consistency resembles soft living tissues, this kind of hydrogel might interest the scientific and medical fields for its ability to trap water and solutes, forming a solid gel structure. Thus, the blending of polycationic and polyanionic biopolymers must be carried out in a water solution, where the pH becomes a key factor as the different charged groups can be more or less ionized, and their interactions affect the physical properties of the matrix [17]. As γ-PGA pKa is about 4.86 and CH pKas are 6.2 and 7.0 [11,17] a pH of 3.5 was optimal for the blending, as at this value, both biopolymers were charged and, consequently, rapidly bind to each other through strong electrostatic interactions and lead to the precipitation of the supramolecular complex. It is worth noting that the variation in CH/γ-PGA ratios and pH values made it impossible to obtain any precipitated material. The resulting glass-like materials were characterized in terms of their solubility both in water and in several organic solvents as well as under different stressing temperatures (70 °C) pH (3.5, 7.0, and 10.0) and mechano-physic (sonication, stirring) conditions. All the glass-like materials tested were insoluble, thus highlighting that, during the ion complex forming and dehydration process, very strong physical crosslinking occurred among the biopolymer chains. It was shown that after 24 h of different solvents, the material samples preserved their physical integrity. In addition, it was observed that the glass-like materials immersed in water rehydrated, whereas those immersed in organic solvents kept their toughness and a glass-like structure, thereby confirming the physical strength of the three-dimensional network that developed [18]. Only hydrogen peroxide at pH 4.0 and 10.0 was found to disintegrate the blended materials due to CH precipitation at a very basic pH and oxidative degradation of polyamide [19] (Figure 2). The pioneering study by Fuoss and Sadek in 1949 explored the interaction between a strongly acidic polyanion (sodium polystyrene sulfonate) and a strongly basic polycation (poly (vinyl methyl pyridinium) chloride) in an aqueous solution [20]. They observed that this interaction led to a colloidal precipitate and the formation of a stable, infusible, and insoluble precipitate, and the reaction occurred rapidly, resulting in a product with nearly stoichiometric equivalents of the component polyions. Shamoun et al. (2012) further investigated stoichiometric polyelectrolyte complexes (PECs) [21] and introduced the term “saloplastics” to describe the materials formed by combining polycations and polyanions, and they demonstrated the compaction of PECs using ultracentrifugation at room temperature with the addition of extensive salt. The study demonstrated the successful creation of stoichiometric PECs in various shapes using an extruder [21]. 

In light of these findings, our materials can be described as thermoplastic and saloplastic, signifying vast applications in the field of PECs, including the creation of 3D forms, PEC micro-chambers (opening up new possibilities for innovative uses), and further explorations in the realm of polyelectrolyte complexes [22,23].

### 2.3. Structural Analyses and Dynamic Mechanical Analysis

Figure 3a,b show Fourier transform infrared (FTIR) spectra of crosslinked materials (CHR1 and CHR2) and neat biopolymers of CH, R1, and R2. The characteristic peaks of neat CH appeared at 3439, 1659, 1550, 1320, 1593, 1420, and 1378 cm^−1^ corresponding to the stretching vibration of N-H and O-H groups engaged in intramolecular and intermolecular hydrogen bonds, C=O stretching of amide I, N-H bending of amide II, C-N stretching of amide III, N-H bending of the primary amine, CH_2_ bending, and CH_3_ symmetrical deformations [19], respectively. Finally, the absorption band at 1154 cm^−1^ can be attributed to asymmetric stretching of the C-O-C glycosidic bridge bond [24]. Regarding R1 and R2, the characteristic peaks observed at 3400, 1727, 1645, 1557, and 1410 cm^−1^ correspond to N–H bonds, C=O stretching vibration of the carboxylic acid, C=O of amide I, C–N stretching modes vibration, and CH_2_ bending deformations, respectively. Notably, the prominent peaks at 1129 cm^−1^ could be related to the melanoidin C-O bending bands [25]. The complexation between the two biopolymers was investigated by comparing the infrared spectra of CHR1 and CHR2 with the neat biopolymers (CH, R1, and R2). Results referred to only physical interactions, as no chemical reactions occurred during the formation of the supramolecular complex [14]. The first evidence of CH complexation with either R1 or R2 was found in the reduction of the intensity of the γ-PGA carboxyl group at 1726 cm^−1^, following its electrostatic and hydrogen bonding engagements with the polar residues of CH (Figure 3). Furthermore, the γ-PGA amide I peak at around 1645 cm^−1^, both in CHR1 (Figure 3a) and in CHR2 (Figure 3b) blends, shifted towards a lower frequency with the increase in CH mass percentage, which evidenced the even stronger physical interaction between the polar residues of the two biopolymers through hydrogen bonding. It has been widely reported in the literature that hydrogen bonding reduces the vibration frequency energy of functional groups engaged in that kind of interaction [26]. The same trend was observed for CH stretching vibration at around 1420 cm^−1^ attributed to the NH_3_^+^ of CH complexing with -COO^−^ of γ-PGA [27]. The FTIR study confirmed successful CH/γ-PGA complex formation via electrostatic attractions between γ-PGA negative charged polar groups and CH protonated amine residues as well as hydrogen bonding interactions.

X-ray diffraction (XRD) profiles of CH, R1, and R2 are shown in Figure 3c. CH diffraction peaks at 2θ 10.5° and 20.5° were recorded. XRD patterns indicated the presence of CH crystallinity structure, whereas both R1 and R2 showed a typical amorphous pattern characterized by a broad amorphous halo centered at 2θ~22°. XRD profiles of CHR1 or CHR2 blends (Figure 3c) indicated amorphous characteristics of the supramolecular complexes, since only amorphous haloes at 2θ~22° and a shoulder at 2θ~40° were visible [28,29]. 

Far-UV circular dichroism (CD) spectroscopy was used to investigate CH, R1, and R2 features in solutions. CH exhibited a spectrum (Appendix A Figure A1) with a deep minimum at 211 nm, indicative of an acetylated amine chromophore on a glycosidic ring in an acidic solution. This signal is related to the acetyl content of the CH, and it is independent of the α, β anomeric equilibrium, the chain length beyond two residues, the ionic strength and pH, and the polymer conformation [30]. R1 and R2 fractions have comparable spectra, showing a deep minimum at about 218 nm when analyzed at pH 6.0, indicating a predominant presence of β-sheet structures in both samples. Decreasing the pH to 3.5 leads to a shift towards 215 nm, with a significant intensity increase (Appendix A Figure A1). This change is typically associated with β-sheets, but the change of intensity associated with the pH change suggests that conformational changes occurred, and a higher content of alpha-helices can be hypothesized considering that the CD spectra of alpha helices are much more intense than those of β-sheets. The limited purity of R1 and R2 prevented the exact determination of secondary structure content via spectrum deconvolution. It is noteworthy that in no case does γ-PGA adopt a random coil conformation. This finding confirms the exclusive presence of γ-PGA in both samples. Its α-helix content increased as pH decreased from 7.0 to 3.0, while β-sheet and random coil contents decreased. On the other hand, α-PGA is mostly a random coil at pH ≥ 6 and α-helical in strongly acid conditions [31]. Furthermore, to streamline the study and gain valuable insights into the behavior of the polymer blends, we focused on one specific type of blended material (CHR1, 1/9) for CD spectra recording. Additionally, DMA analysis was performed on a dry sample CHR1 (3/7) as a representative example for all of the other prepared blended ratios. The CD spectra revealed an intense minimum intermediate between CH and γ-PGA, suggesting that the interaction between the two polymers does not cause significant conformational changes (Figure 3d). 

The DMA analysis of the dry sample CHR1 (3/7) indicated the formation of a crystalline phase after water removal. The storage modulus exhibited a smooth and wide drop, indicating the presence of crystalline fractions relaxing at different times (Figure 3e). Notably, the material demonstrated a high glassy storage modulus, surpassing analogous CH/γ-PGA materials produced through extrusion and the addition of additives [32]. It also showed comparable properties to pristine γ-PGA [33] or CH [34]. Moreover, the glass transition temperature (Tg) of 84.4 °C was higher than the Tg of the reference material [32].

According to the XRD and DMA results, the materials studied exhibit both crystalline and amorphous phases. These findings align with the mechanical model proposed by Takayanagi, which supports the viscoelastic behavior of crystalline polymers by connecting the crystalline and amorphous regions in series and parallel elements [35].

### 2.4. Thermal Analyses

Thermogravimetric (TG) and differential thermogravimetric (DTG) analyses of CH, R1, and R2 were carried out and the obtained curves were normalized to the initial sample size (Figure 4a–d). The TG results showed that weakly bound water evaporated from 50 °C up to about 200 °C [36]. CH lost 25% of water while R1 and R2 15%, confirming the obtained moisture uptake and the content data. Glass-like materials showed minimal water loss due to their crosslinked structures. The absence of very low intense decomposition peaks in the region of water evolution in the differential thermograms was consistent with the findings discussed earlier regarding glass-like materials. The decomposition temperatures of CH (215.0 °C), R1 (257.7–306.0 °C), and R2 (291.1 °C) were aligned with the previous thermal characterization of the same biopolymers [27]. The depolymerization of γ-PGA started at around 200 °C, according to an unzipping cyclo-depolymerization mechanism generating pyroglutamic acid and methyl pyroglutamate, with the decomposition of the ionic complex [11]. Both R1 and R2 showed greater stability than CH as their main polymer degradation occurred at higher temperatures. R1 showed two thermal degradation peaks, likely associated with the thermal depolymerization of γ-PGA with different Mw. TG and DTG curves obtained from the analyses of CHR1 and CHR2 blends prepared at different CH/γ-PGA ratios are also shown in Figure 4a–d. The main polymer degradation phenomena of all the blends were faster than those observed with R1 and R2 but slower than that of CH. These findings highlighted the strong physical links occurring inside the supramolecular complexes due to both electrostatic interactions between the amino groups of CH and the carboxyl group of γ-PGA as well as the intermolecular hydrogen bonding [37]. Although γ-PGA represents the main polymer matrix of the crosslinked materials, the thermal degradation was significantly influenced by the presence of even low amounts of CH, as shown by the shifting of the observed T_onset_ towards lower temperatures. The peaks corresponding to the maximum degradation rate of the polymers (T_max_) overlapped at the value of CH T_max_. This could be due to the higher Mw of CH and to its reactive protonated amine groups being able to strongly attract the negatively charged γ-PGA chains [38]. The lower Mw of γ-PGA contained in the R2 fraction could be responsible for different structural networks, as shown by the presence of at least three degradation kinetics [10]. The last decomposition step observed between 400–500 °C, particularly marked in R2-based materials, was likely associated with the decarboxylation process leading to unsaturated chain fragments and non-volatile compounds in an inert atmosphere (nitrogen).

### 2.5. Morphological Analysis and Disintegration Test

Scanning electron microscope (SEM) analyses of CH, R1, and R2 powders revealed a heterogeneous morphology for biopolymers with irregular particles (100–200 mm) (Figure 5a). As far as R1, elongated/fibrillar structures (200–300 mm) were also present. In addition, SEM images showed a flat and homogeneous structure for all of the glass-like materials prepared at all CH/R1 and CH/R2 ratios (Figure 5b). The observed structure demonstrated that complexation between the oppositely charged biopolymers produced a smoothed surface as well as the absence of voids and/or other morphological defects. Moreover, SEM images confirmed the absence of phase separation between CH and R1 or R2 in all of the blend ratios. Finally, Figure 5c-1,c-2 shows some preliminary results on the disintegration of the glass-like materials. The results show marked changes in the weight over time when samples were buried in the soil [39]. It is well known that as water diffuses from the soil into the polymer samples, their swelling and degradation increases because of the role played by water as a carrier of soil microorganisms able to depolymerize and transform the biopolymers in biomass, CO_2_, and water [40]. All of the sample weights gradually decreased during the experiment. The samples showed low disintegration after 5 days with a loss of weight of 3.0–8.0%, while the degradability increased after 15 days as the mass reached 25.0%. Finally, the degradability further increased in the next 25 days, and the weight loss of the samples was between 13.0 and 85.0% and, in particular, was 62.0% for CHR1 (3/7 ratio) and even higher (85.0%) for CHR2 (3/7 ratio). These results suggest that these materials are degradable in soil under open-air environmental conditions. 

## 3. Materials and Method 

### 3.1. Materials and Chemicals

A not pure poly-γ-glutamic acid (COM-PGA, 50% purity) bulk, produced by microbial fermentation, was purchased by Xi’an Fengzu Biological Technology Co., Ltd. (Xi’an City, Shaanxi Province, China). Chitosan (CH) (75–85% deacetylated chitin, poly-D-glucosamine, 50,000–190,000 Da) was purchased by Sigma Aldrich, St. Louis, MO, USA, hydrogen peroxide (30% H_2_O_2_), acetic acid 96%, toluene, dimethyl sulfoxide 99.9% (DMSO) methanol, 99.9%, and chloroform,99.9% was purchased by CARLO ERBA (Emmendingen, Germany). All of the reagents used to prepare the buffers and the standards employed in UHPLC and SEC-TDA analyses were purchased from Sigma-Aldrich, St. Louis, MO, USA. 

### 3.2. Fractionation of Commercial Poly-γ-Glutamic Acid by Membrane-Based Ultra-Filtration

Two γ-PGA fractions were prepared by UF from the COM-PGA, as previously reported [10]. Briefly, the COM-PGA sample (300 g) was dissolved in MilliQ water (10.0 L), stirred at 600 rpm for 18 h, and then centrifuged at 6500 rpm (4 °C) for 20 min (Avanti J-20XP, Beckman Coulter, USA) to remove insoluble material. The dissolved, centrifuged sample was micro-filtrated with a 0.65 μm polypropylene membrane (total filtering area of 0.05 m^2^, Sartopure PP2 MidiCaps, Sartorius Group, Germany) and a non-automatic Sartoflow Alpha system (Sartorius Group, Göttingen, Germany). After this step, it was ultra-filtered and diafiltered with two volumes of MilliQ water by using two 100 kDa cut-off polyethersulfone cassette membranes (total filtering area of 0.1 m^2^, Sartorius Group, Germany) and an automatic tangential flow filtration system (Uniflux 10, UNICORN, GE Healthcare, Chicago, IL, USA). The 100 kDa retentate sample (R1) was collected while the 100 kDa permeate (P1) was further ultra-filtered and diafiltered with two volumes of MilliQ water by using two 3 kDa cut-off polyethersulfone cassette membranes (0.5 m^2^ of total filtering area, Sartorius Group, Germany) and the abovementioned system. The 3 kDa retentate sample (R2) was recovered while the 3 kDa permeate (P2) was discarded. During the process, samples were taken to determine the dry weight and the γ-PGA concentration. Data on pressure, volumes, conductivity, and the pH of the samples were also collected during the processes.

### 3.3. Determination of the Dry Weight and Water Content of the Samples

Aliquots (10 mL) of the UF samples were freeze-dried (LIO 5P, 5Pascal, Italy, Milan) (18 h at −20 °C and 1.05 mbar, and then 3 h at 20 °C and 0.04 mbar) [41] and then weighed to determine their dry weight. The water content of CH, COM-PGA, R1, and R2 was also determined as previously described [10]. 

### 3.4. Poly-γ-Glutamic Acid Quantitative Determination by Ultra-High Performance Liquid Chromatography

Analyses of the γ-PGA quantity present in the initial COM-PGA sample and in R1 and R2 were performed by UHPLC (Ultimate 3000, Dionex, Sunnyval, CA, USA) using a previously reported method [10]. Briefly, the samples were first hydrolyzed at 2.5 g·L^−1^ with 5 M HCl at 100 °C and 600 rpm for 6 h (Thermomixer comfort, Eppendorf, Germany) and then analyzed by using an ion-exclusion column (Rezex ROA-organic acid H^+^, 300 × 78 mm, Phenomenex, Torrance, CA, USA) and an ultra-high-performance chromatographer (Ultimate 3000, Dionex, Sunnyvale, CA, USA), injecting 10 μL and eluting with 0.1% H_2_SO_4_ for 25 min at 0.8 mL·min^−1^ at 40 °C and detecting at 200 nm. The determined γ-PGA concentrations were then multiplied for the volumes of the samples to calculate the total amount (g) of the polymer in each fraction. These amount values were then divided by the polymer content in the initial COM-PGA sample to calculate the percentage of γ-PGA recovery in the UF process according to Equation (1):% γ-PGA recovery/COM-PGA sample = % [γ-PGA sample (g)/COM-PGA sample (g)] × 100 (1)

Furthermore, in all of the samples, the percentage of the γ-PGA amount concerning the sample dry weight was calculated according to Equation (2):% γ-PGA/dry weight = % (γ-PGA sample (g)/dry weight sample (g)] × 100 (2)

### 3.5. Chitosan and Poly-γ-Glutamic Acid Molecular Weight Determination by Size Exclusion Chromatography with Triple Detector Array

Mw analyses of CH, COM-PGA, and R1 and R2 were performed by using two gel-permeation columns (TSK-GEL GMPWXL, 7.8 × 30.0 cm, Tosoh Bioscience, Italy), put in series in a SEC-TDA instrument (Viscotek, Malvern, Italy), and equipped with a triple detector array with a refractive index (RI), a four-bridge viscosimeter (VIS), and two laser detectors of right-angle (RALS) and low-angle light scattering (LALS). The calibration of the instrument was performed with polyethylene oxide (PEO) standard (22 kDa PolyCAL, Viscotek, Malvern, Italy). CH and γ-PGA were analyzed by using two different methods. CH analyses were run using a buffer made of 0.15 M acetic acid, 0.1 M sodium acetate, and 0.4 mM sodium azide at pH 4.5 and 40 °C with a flow rate of 0.6 mL·min^−1^, as previously described (Nguyen, Winnik, and Buschmann, 2009) [42]. γ-PGA was eluted with 0.1 M sodium nitrate at pH 7.0 and 40 °C, with a flow rate of 0.6 mL·min^−1^, as previously described [43]. The CH and γ-PGA average Mw values, their dispersity (Mw/Mn), and their intrinsic viscosity (IV) were determined in duplicate on the base of the detector signals and the specific dn∙dc^−1^ values by applying Equations (3)–(5), as reported by the manufacturer:RI signal = kRI · dn∙dc^−1^
(3)
VIS signal = kVIS · IV · C (4)
LALS signal = kLALS · Mw · (dn∙dc^−1^ )^2^ ·C (5)
where IV is the intrinsic viscosity (dl × g^−1^), C is the concentration (mg∙mL^−1^), dn∙dc^−1^ is the refractive index increment (mL × g^−1^), and kRI, kVIS, and kLALS are instrumental constants obtained by the universal calibration with PEO (Viscotek; information available from http://www.viscotek.com (accessed on 2 May 2023) [44]. The γ-PGA dn∙dc^−1^ value used for the analyses was 0.183 mL·g^−1^, as experimentally determined in a previous paper [10]. The CH dn∙dc^−1^ value was 0.192 mL·g^−1^, as reported in the literature [43]. In the analyses, the representativity of each peak was calculated as the percentage ratio of the RI area of a single peak divided by the sum of the RI areas of all the peaks in the chromatogram [41]. 

### 3.6. Preparation of Chitosan and Poly-γ-Glutamic Acid Crosslinked Materials (CH/γ-PGA)

First, aqueous solutions of CH (20 mg/mL *w*/*v* in 0.1 M HCl) and the γ-PGA fractions (R1 and R2, 50 mg/mL) were prepared at pH 3.5. Then, R1 and/or R2 solutions were added to CH solutions in different mass ratios (1/9, 2/8, and 3/7; *w*/*w*, CH/R1 or CH/R2), and the resulting mixtures were adjusted at pH 3.5 and stirred at 800 rpm at 25 °C for 60 min. This addition led precipitates to form, which were collected first by decantation and then by centrifugation at 10,000 rpm for 10 min at 4 °C (Avanti J-20 XP, Beckman Coulter, Brea, CA, USA) after being washed three times with distilled water at pH 3.5. The collected materials were in a dried state for characterization. In addition, separate aqueous solutions of R1, R2, and CH at pH 3.5 were lyophilized (Thermo Savant Modulyo Benchtop, USA) in order to characterize the neat polymers. 

### 3.7. Hydrophilicity and Solubility

Solubility qualitative studies were carried out on glass-like materials focusing on the sample CHR1 (2/8 ratio) as a representative material. Briefly, about 10 mg of glass-like materials were immersed for 24 h in 600 µL of water at different pHs (3.5, 7.0, and 10.0) as well as in 600 µL of different solvents at pH 4.0 and pH 10.0 (in 30% hydrogen peroxide (*w*/*v*), chloroform, DMSO, acetic acid, methanol, and toluene). The samples were also further treated by ultrasonication (Bandelin SONOPULS ultrasonic homogenizers, Binder, Tuttlingen, Germany) followed by stirring for 2 h at 70 °C in a water bath to check the insolubility. 

### 3.8. Structural Analyses


*Fourier Transform Infrared Spectroscopy Analysis (FTIR)*


Spectra were recorded by an FTIR instrument (NICOLET 5700, Australia) and 64 scans interferogram was collected with a variable path length cell and KBr windows. 1 mg of CH, R1, and R2 samples, and of the CHR1 and CHR2 blends were ground into powder using Ball Mills (Retch-PM 100 CM, UK Ltd., Hope Valley, UK) combined with 100 mg dry KBr. The grounded mixture was then pressed into a transparent disc and dried at 60 °C for 24 h before analysis. The spectra were recorded at a straight baseline of 400–4000 cm^−1^. 


*X-ray Diffraction Analysis*


Wide angle X-ray scattering measurements (WAXS) of CH, R1, and R2 samples as well as the CHR1 and CHR2 blends were obtained using nickel-filtered CuKα radiation with a Philips automatic diffractometer (Empyrean by Panalytical, Monza, Italy). 


*Circular Dichroism*


CD spectra of CH, R1 and R2, as well as that of the CHR1 (1/9) blend, were recorded at 20 °C using a Jasco J-1500 spectropolarimeter equipped with a Peltier thermostatic cell holder (Model PTC-348WI). CD measurements were carried out in the 250−200 nm range, using a 0.1 cm path length cell. A total of 0.1 mg mL^−1^ CH solutions at pH 3.5 of 0.5 mg mL^−1^ of R1 and R2 solutions, either at pH 6.0 or 3.5 and 0.5 mg mL^−1^ CHR1 (1/9 ratio) blend suspension at pH 3.5, were analyzed with 0.5 nm data pitch, 2 nm bandwidth, and 20 nm min^−1^ scanning speed. Each spectrum was obtained as the average of three measurements and corrected for solvent signal.

### 3.9. Dynamic Mechanical Properties Analysis (DMA)

The mechanical properties of the blends were evaluated by dynamic mechanical analysis (DMA) performed with a Tritec 2000 DMA (Triton technology, London, UK). Hydrogels of CHR1 (3/7) were dried and formed in the shape of bars with a rectangular section (10 × 1 mm^2^). Tests were conducted at 1 Hz with a displacement of 0.01 mm and heated at 2 °C min^−1^ starting from room temperature.

### 3.10. Thermal Analyses

Thermogravimetric analyses of CH, R1, and R2, as well as of CHR1 and CHR2 glass-like materials, were performed using a thermogravimetric analyzer (TGA/DTA, Perkinelmer pyrisdiamond) equipped with a gas station. An amount of 3–4 mg of each sample was placed in an open ceramic crucible and heated from 25 °C up to 600 °C at a speed rate of 10 °C/min under a nitrogen flow of 30 mL/min. Before testing, the samples were conditioned for 24 h at 25 °C and 50% RH. 

### 3.11. Scanning Electron Microscopy

The morphological structure of the CH, R1, and R2 powders, as well as the cross-section fracture of CHR1 and CHR2 glass-like materials, were analyzed by SEM (Nova NanoSem 450-FEI-Thermo Fisher, Scientific, Waltham, MA, USA). The cryo-fractured samples were coated with thin layers of gold/palladium alloy using a sputter, and the images were then taken at an accelerating voltage of 5 kV.

### 3.12. Disintegration Test

All of the glass-like materials were dried at 105 °C, weighed, and buried in agricultural clay soil at the Faculty of Agriculture and Veterinary Medicine farm at An Najah National University in Tulkarm, Palestine. Properly labeled steel meshes were used to insert the samples inside the soil in 15 cm deep holes and to recollect them more easily in order to measure the weight loss. Before and after burying the samples, the soil was sifted to remove large clumps and plant debris. The soil moisture was controlled using a moisture indicator (IRROMETER. REG. US PAT. OFF) and kept at about 20% during the study by adding water whenever necessary. The buried samples were dug out once every 10 days, washed in distilled water, dried in the oven at 105 ± 2 °C for 6 h, and equilibrated in a desiccator for at least one day. The samples were then weighed, and the weight loss (%) was expressed according to Equation (6):Weight loss (%) = Degraded sample weight(g)/Initial sample weight × 100 (6)

### 3.13. Statistical Analysis

All the results are presented as mean ± standard deviations of the three replicates. Statistical analyses were performed with Microsoft Excel software (Microsoft Office 2017). The obtained data were tested for significant differences through the *t*-test, and data with *p* < 0.05 were considered to have statistical significance.

## 4. Conclusions

Advanced bio-composites made of CH and low Mw γ-PGA fractions were prepared by a process based on mixing distinct solutions of the two biopolymers at specific ratios and at acidic pH. The sedimented supramolecular complexes appeared as rubbery hydrogels that, when dehydrated, gave rise to glass-like materials endowed with advantageous physical characteristics and thermal stability as well as the amorphous and crystalline structure of the obtained glass-like material. This opens a window for the potential use of these new bio-composites in diverse fields. 

Their applications could span regenerative medicine, biomedical devices, food packaging, and even 3D printing. The non-toxicity of the constituent biopolymers further enhances their suitability for these applications [45,46,47]. Due to its protein-like effects, this polymer composite holds significant potential, and may even outperform conventional polymers, thereby making it suitable for particular applications in regenerative medicine, such as the restoration of segmental bone defects [48].

In summary, the successful development of these advanced bio-composites opens up vast possibilities and represents a promising step towards sustainable and innovative materials for a wide range of applications in different industries. Further exploration and refinement of these materials will undoubtedly unveil even more exciting opportunities for their utilization in various fields.

## Figures and Tables

**Figure 1 ijms-24-12495-f001:**
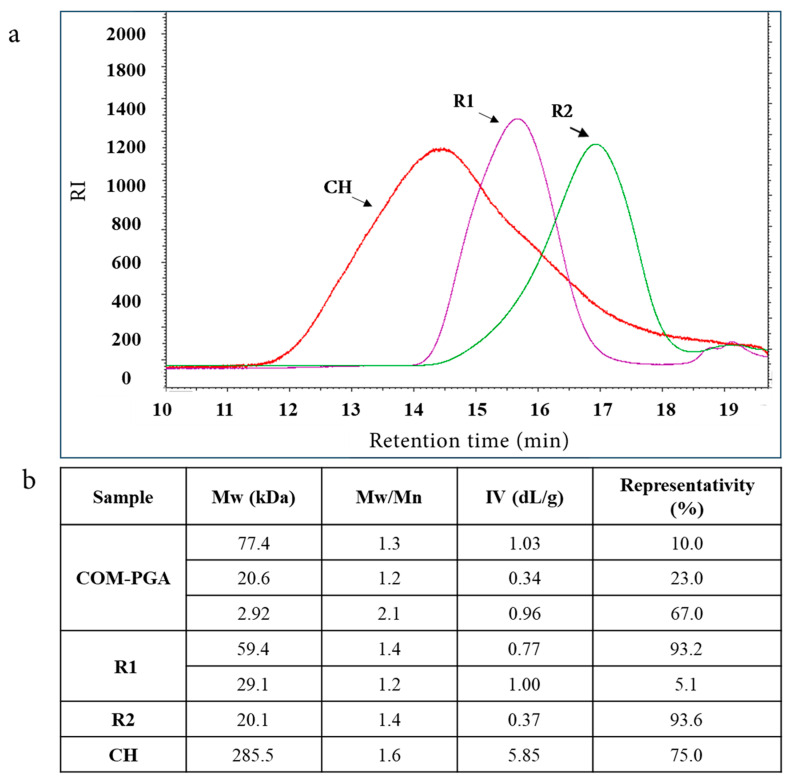
(**a**) Overlay chromatograms of SEC-TDA analyses of chitosan (CH), R1, and R2 ultrafiltrated fractions of poly-γ-glutamic acid (COM-PGA). (**b**) Average molecular weight (Mw), dispersity (Mw/Mn), intrinsic viscosity (IV), and representativity of the main peaks.

**Figure 2 ijms-24-12495-f002:**
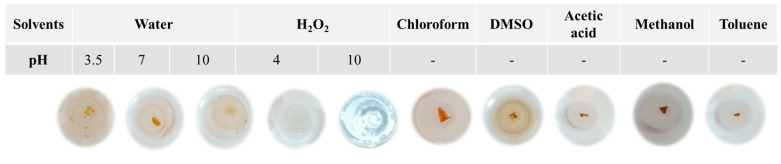
Images of CHR1(2/8) sample immersed in distilled water at different pH values (3.5, 7.0, and 10.0), or in 30% (*w*/*v*) of H_2_O_2_ at different pH 4.0 and 10.0, or in different organic solvents.

**Figure 3 ijms-24-12495-f003:**
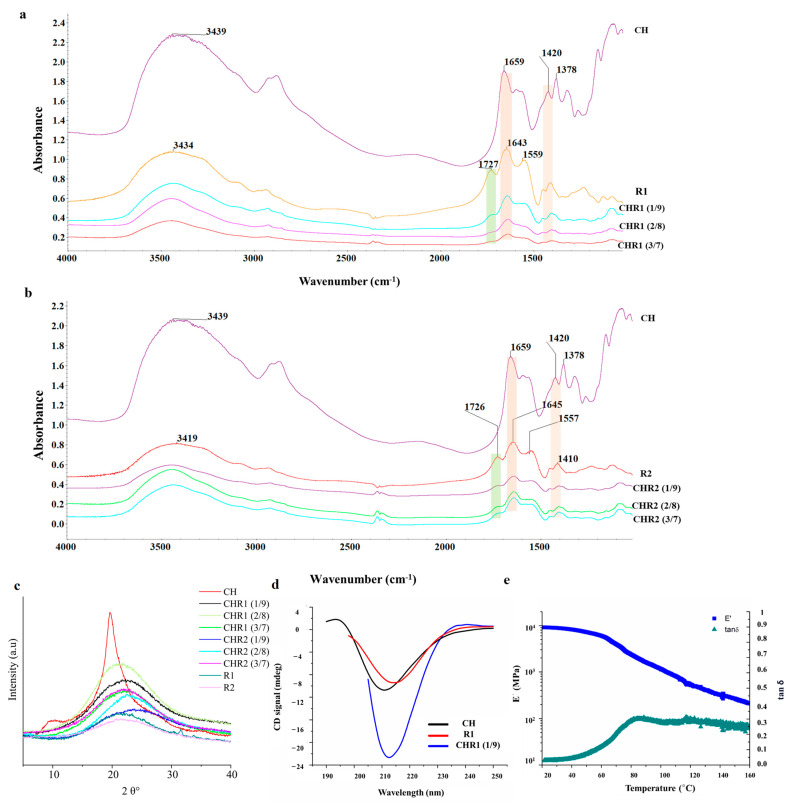
(**a**) Fourier transform infrared spectroscopy (FTIR) spectra of CHR1. (**b**) FTIR spectra of CHR2 blends. (**c**) X-ray diffractograms of CHR1 and CHR2 using three different ratios CH/R1 or CH/R2 (1/9, 2/8 and 3/7) in comparison with the single biopolymers of chitosan (CH) and poly-γ-glutamic acid (γ-PGA) fractions (R1 and R2). (**d**) Circular dichroism spectra of CHR1 (1/9 ratio) in comparison with CH and γ-PGA R1 fraction. (**e**) Storage modulus (E′) and tan δ values by DMA analysis of the crosslinked materials of CHR1 (3/7).

**Figure 4 ijms-24-12495-f004:**
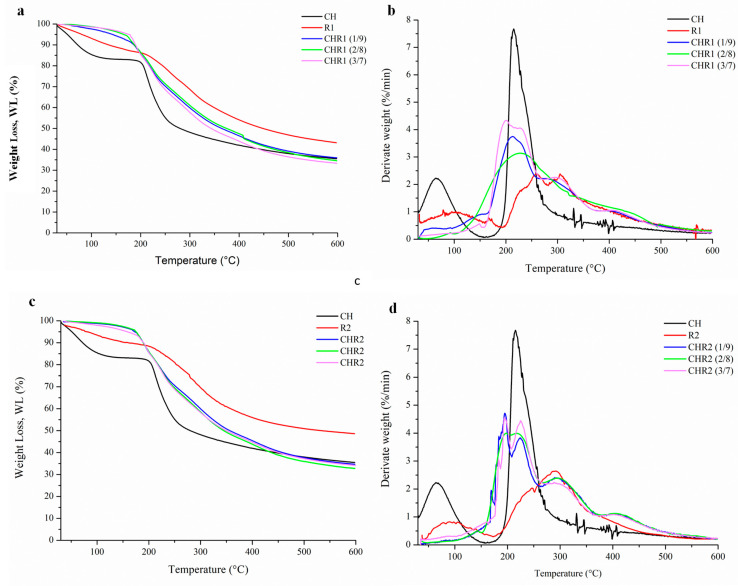
Thermogravimetric (TG) (**a**,**c**) and differential thermogravimetric (DTG) (**b**,**d**) analyses of CHR1 and CHR2 blends, obtained with three different CH/R1 or CH/R2 ratios (1/9, 2/8 and 3/7). The curves were compared to the single biopolymers of chitosan (CH) and poly-γ-glutamic acid fractions (R1 and R2).

**Figure 5 ijms-24-12495-f005:**
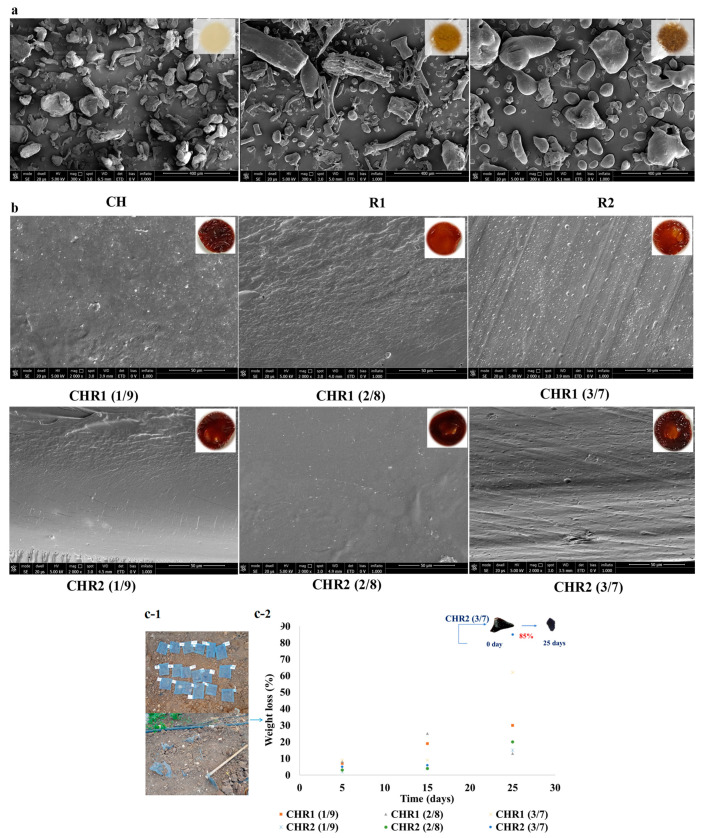
(**a**) Scanning electron microscopy (SEM) images of chitosan (CH) and of poly-γ-glutamic acid fractions (R1 and R2) captured at magnitude 2000× (the scale bar was 400 μm); the inserts represent the visual aspect of the three analysed powders. (**b**) SEM images (cross-section) of glass-like materials (CHR1, upper panels; CHR2, lower panels) obtained using the three different CH/R1 or CH/R2 ratios (1/9, 2/8, and 3/7) at magnitude 2000× (scale bar was 50 μm). The inserts represent the visual aspect of the three analysed glass-like materials. (**c-1**) Pictures of the blended materials buried in the soil. (**c-2**) Degree of disintegration, expressed as % weight loss, of CHR1 and CHR2 blends obtained at three different CH/R1 or CH/R2 ratios (1/9, 2/8, and 3/7) buried in the soil (orange square for CHR1 (1/9), grey triangle for CHR1 (2/8), yellow cross for CHR1 (3/7); blue cross for CHR2 (1/9), green circle for CHR2 (2/8), blue circle for CHR2 (3/7). Blended material at the beginning (T0) and after 25 days (T25) of degradation is shown in the insert of the panel.

## Data Availability

Not applicable.

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
