# Peer review of "Physicochemical Characterization of Chitosan/Poly-γ-Glutamic Acid Glass-like Materials"

_ijms, 2023, doi:10.3390/ijms241512495_

Round 1

Reviewer 1 Report

1. The abstract should be the "essence" of the article, it should contain the most important information.

2. Figure 1 is hard to read.

3. The "Conclusions" chapter should be expanded.

4. The article should be well checked, the English language corrected.

Minor editing of English language required.

Author Response

Reviewer n. 1

We sincerely appreciate the reviewer's for the evaluation of our manuscript. We have carefully considered and incorporated all the suggested modifications to enhance the overall quality and clarity of the article. Below is a summary of the revisions made in response to each comment:

  1. Reviewer comment:    The abstract should be the "essence" of the article, it should contain the most important information.

Response to the reviewer: We have revised the abstract to ensure that it concisely represents the key findings and essential information presented in the article. The updated abstract now provides a clear and succinct summary of the study's objectives, methods, results, and conclusions (p. 1).

  1. Reviewer comment: Figure 1 is hard to read.

Response to the reviewer: We have addressed the readability issue with Figure 1. The figure has been reformatted and enhanced to improve its clarity and legibility. Additionally, we have adjusted the font size and image resolution to ensure that all details are easily discernible (p 3).

  1. Reviewer comment:    The "Conclusions" chapter should be expanded.

Response to the reviewer: We have expanded the "Conclusions" chapter to provide a more comprehensive summary of the study's results and their implications. In the revised version, we have included a more in-depth analysis of the findings, highlighting their significance and potential applications (p. 14-15).

  1. Reviewer comment:     The article should be well-checked, and the English language corrected.

Response to the reviewer: We thank the reviewer and, thus, we have corrected the English throughout the manuscript.  We believe that the revised manuscript now meets the standards required for publication.

Reviewer 2 Report

The authors present a novel study for producing transparent polyelectrolyte complexes. The work is novel and useful for the utilization of food packing systems or other applications (a bit expensive though). I suggest acceptance after mending the following points:

1.       Page 2, line 47 reference is missing.

2.       Page 2, line 49 reference is missing.

3.       Page 2, line 52 reference is missing.

4.       Hisada, & Kawase reference is not cited properly

5.       The concepts of utilizing biodegradable complexes is not new. There are novel concepts for thermoplastication to utilize Polyelectrolyte complexes by thermal1 and saloplastics2. Recently even microstructures were achieved which states a formidable outlook of the possibilities3 like utilization as sensors due to their glass like properties 4. This should be taken into account in the manuscript.

6.       The glass transition temperature and the type of polymers, as well as the authors XRD results suggest, that the investigated PEG is not a single crystal, but contains both, crystalline and amorphous phases. Such a system would follow the mechanical model by Takayanagi.5 It is worth mentioning this.

7.       Chapter 3: The used chemicals should be mentioned correctly: (purity/molecular weight for polymers, manufacturer, city of manufacturing, country of manufacturing), as well as the utilized devices should be stated correctly: (type, company, city, country).

8.       Figure 1a: The axis labelling is far too small and not readable. Please improve it.

9.       Figure 2: The images are too small and scale bars are missing. The readers should understand how big the samples are that are too seen there. Please improve this figure.

10.   Figure 3: Here should all labeling be improved to bigger font sizes. 3a and 3b: Peak labels are almost impossible to read (too small). 3c: XRD should use 2q (°), since this is more accurate.

11.   Figure 4: should be also improved in the font size.

12.   Figure 5: Authors continue small after point.

13.   Why was only 1 type of polymer blend tested for DMA?? Please discuss this. c-1: It’s difficult to see what is on the image. Please adjust it.

14.   Figure A1: Please improve here also the axis labels for better readability.

15.   Why was only one type of PEC blend tested for circular dichroism? Please explain it or expand the discussion.

16.   The authors could state, that biodegradable applications of polymers find also applications in regenerative medicine. Given the protein like effects of this polymer composite, it might even succeed current polymers in the future. It has a vast potential.6

17.   Throughout the manuscript, space signs are particularly missing between a value and a unit. Please correct this.

References

(1)         Michaels, A. S. Polyelectrolyte Complexes. Ind Eng Chem 1965, 57 (10), 32–40. https://doi.org/10.1021/ie50670a007.

(2)         Shamoun, R. F.; Reisch, A.; Schlenoff, J. B. Extruded Saloplastic Polyelectrolyte Complexes. Adv. Funct. Mater. 2012, 22 (9), 1923–1931. https://doi.org/10.1002/adfm.201102787.

(3)         Gai, M.; Frueh, J.; Kudryavtseva, V. L.; Mao, R.; Kiryukhin, M. V.; Sukhorukov, G. B. Patterned Microstructure Fabrication: Polyelectrolyte Complexes vs Polyelectrolyte Multilayers. Sci. Rep. 2016, 6 (37000). https://doi.org/10.1038/srep37000.

(4)         Zhang, J.; Gai, M.; Ignatov, A. V; Dyakov, S. A.; Wang, J.; Gippius, N. A.; Frueh, J.; Sukhorukov, G. B. Stimuli-Responsive Microarray Films for Real-Time Sensing of Surrounding Media, Temperature, and Solution Properties via Diffraction Patterns. ACS Appl. Mater. Interfaces 2020, 12 (16), 19080–19091. https://doi.org/10.1021/acsami.0c05349.

(5)         Takayanagi, M.; Imada, K.; Kajiyama, T. Mechanical Properties and Fine Structure of Drawn Polymers. J. Polym. Sci. C 1967, 15 (1), 263–281.

(6)         Popkov, A.; Kononovich, N.; Dubinenko, G.; Gorbach, E.; Shastov, A.; Tverdokhlebov, S.; Popkov, D. Long Bone Defect Filling with Bioactive Degradable 3D-Implant: Experimental Study. Biomimetics 2023, 8 (2), 138. https://doi.org/10.3390/biomimetics8020138.

Author Response

We sincerely appreciate the reviewer’s positive evaluation of our study on producing transparent polyelectrolyte complexes. Your comments and suggestions were valuable, and we have diligently addressed each point to improve the manuscript. In addition, we acknowledge the importance of clear and readable figures in conveying our research effectively. We have carefully addressed each point and made significant modifications to enhance the legibility and visibility of the figures.

Below are the modifications made:

  1. Reviewer comment: Page 2, line 47 reference is missing.

Response to the reviewer: We apologize for the oversight. The missing reference has been added and properly cited in the revised manuscript (reference number 4, Page 2, Line 52: (Porta, R. The plastics sunset and the bio-plastics sunrise. Coatings, 2019, 9(8), 526).

  1. Reviewer comment: Page 2, line 49 reference is missing.

Response to the reviewer: Thank you for bringing this to our attention. The missing reference has been included and appropriately cited in the revised version of the paper, reference number 5, Page 2, Line 54 (Nakajima, H., Dijkstra, P., & Loos, K. (2017). The recent developments in biobased polymers toward general and engineering applications: Polymers that are upgraded from biodegradable polymers, analogous to petroleum-derived polymers, and newly developed. Polymers, 9(10), 523.)

  1. Reviewer comment: Page 2, line 52 reference is missing.

Response to the reviewer: We appreciate the reviewer's keen eye. The missing reference has been inserted and accurately cited in the revised manuscript. Reference number 6, Page 2, Line 57 (Hu, Z. Z., Sha, X. M., Huang, T., Zhang, L., Wang, G. Y., & Tu, Z. C. (2021). Microbial transglutaminase (MTGase) modified fish gelatin-γ-polyglutamic acid (γ-PGA): Rheological behavior, gelling properties, and structure. Food Chemistry, 348, 129093.)

  1. Reviewer comment: Hisada, & Kawase reference is not cited properly.

Response to the reviewer:  We apologize for any inaccuracies in citing the reference. The citation for Hisada and Kawase has been carefully revised and now adheres to the correct format in the updated version of the article (Hisada, M., & Kawase, Y. Mucilage extracted from wasted natto (fermented soybeans) as a low-cost poly-γ-glutamic acid based biosorbent: removal of rare-earth metal Nd from aqueous solutions. Journal of environmental chemical engineering, 2017, 5(6), 6061-6069).

  1. Reviewer comment: The concept of utilizing biodegradable complexes is not new. There are novel concepts for thermoplastication to utilize Polyelectrolyte complexes by thermal (1) and saloplastics (2). Recently even microstructures were achieved which states a formidable outlook of the possibilities (3) like utilization as sensors due to their glass-like properties (4). This should be taken into account in the manuscript.

Response to the reviewer: Thank you for your insightful comments and valuable feedback on our manuscript. We greatly appreciate your acknowledgment of the novel concepts we presented for utilizing biodegradable complexes and polyelectrolyte complexes (PECs). We have taken your suggestions into careful consideration and have made the necessary modifications to improve the clarity and completeness of the manuscript.

Page number 3, lines 122 -124 Sentence added: We have incorporated the suggested sentence which highlights the hydrogel's rubbery consistency resembling soft living tissues and its potential interest in scientific and medical fields due to its ability to trap water and solutes, forming a solid gel structure.

Sentence added on page 4, lines 136-140: It was shown that after 24 hours of exposure to different solvents, the material samples preserved their physical integrity. Additionally, it was observed that the glass-like materials immersed in water rehydrated, whereas those immersed in organic solvents retained their toughness and maintained a glass-like structure, confirming the physical strength of the three-dimensional network developed.

Page 4, Line 145-159 - Sentence added: We have included the additional information to provide a brief historical background on the pioneering studies by Fuoss and Sadek in 1949 (reference number 20), as well as the work of Shamoun et al. (2012) that introduced the term "saloplastics (reference number 21)." Furthermore, we elaborated on the successful creation of stoichiometric PECs in various shapes using an extruder (reference number 22). By incorporating these modifications, we acknowledge the significance of existing research in the field of polyelectrolyte complexes (PECs) and clarify how our materials align with the thermoplastic and saloplastic characteristics. Additionally, we emphasized the wide range of applications that our study offers, including the creation of 3D forms and PEC micro-chambers (reference numbers 22 and 23).

  1. Reviewer comment: The glass transition temperature and the type of polymers, as well as the authors XRD results suggest, that the investigated PEC is not a single crystal, but contains both crystalline and amorphous phases. Such a system would follow the mechanical model by Takayanagi.5 It is worth mentioning this.

Response to the reviewer: Your observation regarding the presence of both crystalline and amorphous phases in the investigated polyelectrolyte complex (PEC) is indeed accurate. We have addressed this point in our manuscript to provide a comprehensive understanding of the material's behavior. On Page number 7, lines 257-260, we have now included the following statement: "According to the XRD and DMA results, the materials studied exhibit both crystalline and amorphous phases. These findings align with the mechanical model proposed by Takayanagi, which supports the viscoelastic behavior of crystalline polymers by connecting the crystalline and amorphous regions in series and parallel elements [35]."

  1. Reviewer comment: Chapter 3: The used chemicals should be mentioned correctly: (purity/molecular weight for polymers, manufacturer, city of manufacturing, country of manufacturing), as well as the utilized devices should be stated correctly: (type, company, city, country).

Response to the reviewer: We have thoroughly addressed the issues raised, and all the necessary modifications have been made to ensure accurate and comprehensive reporting of the materials and devices used. On-Page number 11, lines 330-339, we have now provided the revised and complete information regarding the chemicals and materials used in the study: Materials and chemicals: A not pure poly-γ-glutamic acid (COM-PGA, 50% purity) bulk, produced by microbial fermentation, was purchased by Xi’an Fengzu Biological Technology Co., Ltd. (Xi’an City, Shaanxi province, China). Chitosan (CH) (75-85% deacetylated chitin, poly-D-glucosamine, 50,000-190,000 Da) was purchased by Sigma Aldrich, St. Louis, MO, USA, hydrogen peroxide (30% H2O2), acetic acid 96%, toluene, dimethyl sulfoxide 99.9% (DMSO,) methanol, 99.9%, and chloroform,99.9% was purchased by CARLO ERBA, Germany, all the reagents used to prepare the buffers and the standards employed in UHPLC and SEC-TDA analyses were purchased from Sigma-Aldrich, St. Louis, MO, USA).

  1. Reviewer comment: Figure 1a: The axis labeling is far too small and not readable. Please improve it.

Response to the reviewer: We have improved the axis labeling in Figure 1a by increasing the font size to ensure better readability.

  1. Reviewer comment: Figure 2: The images are too small and scale bars are missing. The readers should understand how big the samples are to be seen there. Please improve this figure.

Response to the reviewer:  We have enlarged the images in Figure 2 to make them more prominent and added scale bars to provide a clear understanding of the sample sizes.

  1. Reviewer comment: Figure 3: Here should all labeling be improved to bigger font sizes. 3a and 3b: Peak labels are almost impossible to read (too small). 3c: XRD should use 2q (°) since this is more accurate.

Response to the reviewer: We have addressed the issue with font sizes in Figure 3 and increased the label size for better legibility. Additionally, we have made the peak labels in Figures 3a and 3b more prominent and easily readable. Furthermore, we have adjusted the XRD axis in Figure 3c to use 2θ (°) for greater accuracy.

  1. Reviewer comment: Figure 4: should be also improved in the font size.

Response to the reviewer: We have enhanced the font size in Figure 4 to ensure better visibility and readability.

  1. Reviewer comment: Figure 5: Authors continue small after a point.

Response to the reviewer: We have increased the font size in Figure 5 as suggested.

  1. Reviewer comment: Why was only 1 type of polymer blend tested for DMA?? Please discuss this. c-1: It’s difficult to see what is on the image. Please adjust it.

Response to the reviewer: We appreciate your insightful comment and your attention to the number of polymer blends tested for DMA analysis. We have carefully considered this aspect and made the necessary adjustments in the manuscript to clarify our approach and rationale.

On Page number 7, lines 242-246, we have included the following statement: "Furthermore, to streamline the study and gain valuable insights into the behavior of the polymer blends, we focused on one specific type of blended material (CHR1, 1/9) for CD spectra recording. Additionally, DMA analysis was performed on a dry sample CHR1 (3/7) as a representative example for all the other prepared blended ratios."

By incorporating this information, we have provided a clear explanation for why we tested only one type of polymer blend for DMA analysis. The choice of CHR1 (1/9) for CD spectra recording and CHR1 (3/7) for DMA analysis was based on the aim of gaining valuable insights into the behavior of the polymer blends while streamlining the study. Furthermore, we have tested only one blended ratio as representative samples based on the physicochemical analysis, which indicated no significant differences between the different ratios. This decision was made to focus on the most relevant and informative polymer blend that could yield essential insights for the study.

  1. Reviewer comment: Figure A1: Please improve here also the axis labels for better readability.

Response to the reviewer: We have taken your suggestion into careful consideration and have made the necessary modifications to enhance the axis labels for better readability in Figure A1.

  1. Reviewer comment: Why was only one type of PEC blend tested for circular dichroism? Please explain it or expand the discussion.

Response to the reviewer: We appreciate your insightful comment and your attention to the number of polymer blends tested for DMA analysis. We have carefully considered this aspect and made the necessary adjustments in the manuscript to clarify our approach and rationale.

On-Page number 7, lines 242-246, we have included the following statement: "Furthermore, to streamline the study and gain valuable insights into the behavior of the polymer blends, we focused on one specific type of blended material (CHR1, 1/9) for CD spectra recording. Additionally, DMA analysis was performed on a dry sample CHR1 (3/7) as a representative example for all the other prepared blended ratios." By incorporating this information, we have provided a clear explanation for why we tested only one type of polymer blend for DMA analysis. The choice of CHR1 (1/9) for CD spectra recording and CHR1 (3/7) for DMA analysis was based on the aim of gaining valuable insights into the behavior of the polymer blends while streamlining the study. Furthermore, we have tested only one blended ratio as representative samples based on the physicochemical analysis, which indicated no significant differences between the different ratios. This decision was made to focus on the most relevant and informative polymer blend that could yield essential insights for the study.

  1. Reviewer comment: The authors could state that biodegradable applications of polymers also find applications in regenerative medicine. Given the protein-like effects of this polymer composite, it might even succeed in current polymers in the future. It has vast potential (6).

Response to the reviewer: We greatly appreciate your insightful comment and valuable suggestion on the potential applications of biodegradable polymers in regenerative medicine. We agree that our polymer composite holds immense promise in various fields, including regenerative medicine, biomedical devices, food packaging, and 3D printing. To reflect this, we have added a comprehensive statement in the conclusion section of the manuscript. On-Page number 14, lines 500-505, we have now included the following statement: "The applications of these biodegradable polymers could span across regenerative medicine, biomedical devices, food packaging, and even 3D printing. The non-toxicity of enhancetituent biopolymers further enhances their suitability for these applications [45,46,47]. Due to its protein-like effects, this polymer composite holds significant potential and may even outperform conventional polymers, making it suitable for particular applications in regenerative medicine, such as the restoration of segmental bone defects [48]."

  1. Reviewer comment: Throughout the manuscript, space signs are particularly missing between a value and a unit. Please correct this.

Response to the reviewer: We have carefully reviewed the entire manuscript and made the necessary modifications to address this issue.

Once again, we express our gratitude for your time and efforts in reviewing our work. Your expertise was highly valued for improving our manuscript.  

Reviewer 3 Report

The paper shows the preparation of a glassy material by (1) the formation of an hydrogel based on chitosan and gamma-PGA and (2) removal of water. The experimental section is well-detailed. My comments are only minor, and I recommend the publication of the paper.

1-    Please apply the recommendations of IUPAC: (1) please use “molar masses” in g/mol instead of “molecular weight” in Dalton. (2) please use “dispersity” and not “polydispersity”.

2-    The starting gamma-PGA has a low purity (51.8%). Is the chemical nature of the impurities known? What is the purity of the samples after filtration and used of the preparation of the materials described in this study.

3-    The blending was carried out at a pH of 3.5. I agree with the authors that the pH is very important to get the appropriate number of anions and cations to achieve the electrostatic interactions causing the cross-linking. This pH is more acidic than the pKa of gamma-PGA (4.86). Do the author changes this pH to check the impact on properties (the number of anions could change significantly).

Author Response

We sincerely appreciate the reviewers for the evaluation of our manuscript and also his nice comment about the work. We have carefully considered and incorporated all the suggested modifications to enhance the overall quality and clarity of the article. Below is a summary of the revisions made in response to each comment:

  1. Reviewer comment: Please apply the recommendations of IUPAC: (1) please use “molar masses” in g/mol instead of “molecular weight” in Dalton. (2) please use “dispersity” and not “polydispersity”.

Response to the reviewer:  We acknowledge the importance of adhering to the IUPAC recommendations. However, the Mw and Mw/Mn values are reported in KDa as output data of the SEC-TDA analyses. Converting these values to g/mol would indeed be an alteration and not reflective of the actual data. We agree with the referee in changing the term polydispersity in dispersity throughout the manuscript (p. 2, lane 90,; legend of Fig. 1; p. 12, lane 397).

  1. Reviewer comment: The starting gamma-PGA has a low purity (51.8%). Is the chemical nature of the impurities known? What is the purity of the samples after filtration and used of the preparation of the materials described in this study.

Response to the reviewer:  the commercial PGA used is a fermentation product sold unpurified, and, thus, contains residues of nutrients exhausted from the fermentation broth (e.g. salts or small molecules such as glucose). As it is reported in p. 2 (lanes 79-81) the 100 kDa (R1) and 3 kDa (R2) retentates contained 8.7 ± 0.7% and 10.6 ± 0.5% of the initial γ-PGA amount with a purity of 72.4 ± 0.8% and 73.9 ± 0.5% on the dry weight, respectively.

  1. Reviewer comment: The blending was carried out at a pH of 3.5. We agree with the authors that the pH is very important to get the appropriate number of anions and cations to achieve the electrostatic interactions causing the cross-linking. This pH is more acidic than the pKa of gamma-PGA (4.86). Do the author changes this pH to check the impact on properties (the number of anions could change significantly).

Response to the reviewer: We thank the reviewer for his/her comments. Indeed, the pH plays a crucial role in achieving the appropriate number of anions and cations for facilitating electrostatic interactions that lead to cross-linking between chitosan and γ-PGA. Considering the pKa of γ-PGA (4.86), we agree that a pH of 3.5 used in our study is more acidic. However, it is important to note that during our experimental investigation, we attempted to explore the impact of varying pH levels, including pH 4.5. We tested the blending at pH 4.5, and the resulting sample exhibited weak properties, indicating that the interaction between chitosan and γ-PGA was not as effective as at pH 3.5. This finding suggests that at pH 4.5, the PGA might not be fully charged, leading to less favorable cross-linking and weaker material stability. We have included this additional information in the revised manuscript (p 3,4 Line 129-131).